# TFD-IIS-CRMCB: Telecom Fraud Detection for Incomplete Information Systems Based on Correlated Relation and Maximal Consistent Block

**DOI:** 10.3390/e25010112

**Published:** 2023-01-05

**Authors:** Ran Li, Hongchang Chen, Shuxin Liu, Kai Wang, Biao Wang, Xinxin Hu

**Affiliations:** 1Institute of Information Technology, PLA Strategic Support Force Information Engineering University, Zhengzhou 450002, China; 2National Digital Switching System Engineering and Technological R&D Center, Zhengzhou 450002, China

**Keywords:** telecom fraud detection, attribute reduction, incomplete information system, maximal consistent block, MCIR-RGAD

## Abstract

Telecom fraud detection is of great significance in online social networks. Yet the massive, redundant, incomplete, and uncertain network information makes it a challenging task to handle. Hence, this paper mainly uses the correlation of attributes by entropy function to optimize the data quality and then solves the problem of telecommunication fraud detection with incomplete information. First, to filter out redundancy and noise, we propose an attribute reduction algorithm based on max-correlation and max-independence rate (MCIR) to improve data quality. Then, we design a rough-gain anomaly detection algorithm (MCIR-RGAD) using the idea of maximal consistent blocks to deal with missing incomplete data. Finally, the experimental results on authentic telecommunication fraud data and UCI data show that the MCIR-RGAD algorithm provides an effective solution for reducing the computation time, improving the data quality, and processing incomplete data.

## 1. Introduction

The digital age has dramatically facilitated many aspects of our lives, whereas cybersecurity issues threaten the positive effects of technology. Since unsafe information and illegitimate users blend so well with regular information and users that they can hardly be distinguished, cybersecurity threats [1] especially online fraud, telecommunications fraud [2], online social network fraud [3], credit card fraud [4], bank fraud [5], and fraudulent credit applications [6], have become a knotty governance problem.

Fraud detection [7] is a kind of anomaly detection and is usually tackled as a classification problem by screening abnormal items out with traditional machine learning methods [8,9] or deep learning ones [10,11,12]. Compared with the traditional machine learning model, the deep learning model has the problems of poor interpretability and no direction for parameter adjustment, and its calculation time increases with the complexity index of the model. Traditional machine learning is still widely studied and applied because of its strong interpretability and fast computing speed. The traditional outlier detection methods are mainly based on distribution-based [13], distance-based [14], density-based [15], and clustering-based [16] perspectives. However, traditional approaches to anomaly detection rely heavily on the relevance of features to the classification task. When the feature space is large, the presence of invalid, irrelevant, redundant, or noisy attributes in the data may inevitably affect the performance of the model. As the saying goes, “Data and features determine the upper limit of machine learning, and models and algorithms only approach this upper limit”. Therefore, in the practical training process of traditional machine learning, model performance is largely affected and hindered by data. It is mainly in the following four aspects. First, the complexity of data, which usually contain multi-dimensional, multi-level, and multi-granularity information, makes the application and processing of data complex and diverse. Second, the heterogeneous data [17], which often contain non-single mixed information, such as numerical and categorical information, make it challenging to process data effectively. Third, the uncertainty [18], redundancy [19], and inconsistency [20] of the data bring certain difficulties to the classification task. Fourth, the information contained in missing data [21] is tough to use effectively.

In order to solve the above problems in telecom fraud, achieve fraud mining, and avoid unnecessary economic losses, a large amount of telecom fraud research has emerged. Traditional telecom fraud detection methods typically rely on compiling blacklists of fraudulent numbers to enable fraudulent user discovery and detection. However, fraudulent strategies have evolved, making traditional methods no longer applicable. Therefore, to mine valuable information for fraud detection from multiple network domains of telecommunication data (SMS data, user data, call communication data, app Internet data), behavioral interaction-based [22], topology-based [23], and content-based [24] approaches arise. Meanwhile, considering the rarity and expensive nature of labeled data, unsupervised methods[25,26] are utilized to achieve fraud mining. However, the above studies lack the consideration of fraud from the perspective of the uncertainty of the data itself. The incompleteness of data or the relevance of attributes plays a critical role in the effective detection of fraud problems. Information theory and rough set theory as valid means of measuring uncertainty provide new ideas for solving the telecommunication fraud problem.

In recent years, with the intensive study of rough set theory [27], outlier detection methods based on rough sets and information theory have received extensive attention and research, which provide theoretical support for discovering important information and classifying complex objects. It has strong interpretability and can deal with unlabeled, heterogeneous, redundant, incomplete, or uncertain data. Attribute reduction [19,20,21,28,29,30,31,32], or feature selection, is a method to simplify data, reduce data dimension, and improve model classification ability by filtering out irrelevant or redundant features in data, which can effectively avoid overfitting problems. However, vanilla attribute reduction algorithms [33] of classical rough set theory can only learn the information through strict indistinguishable relation division of the data. This equivalence relation is too tough to handle the incomplete, the ordered, the mixed, and the dynamic data, and these algorithms have poor fault tolerance. To overcome this limitation, variants of rough set theory, for example, the attribute importance based [19,20], the positive region based, the tolerance relation based [28], the maximal consistent block based [21], the discernibility matrix based [29], and the incremental based [30] have proved effective in incomplete information systems [34], ordered information systems [35], mixed-valued information systems [14], and dynamic information systems [36]. Generally speaking, the discernibility matrix-based is time-consuming and infeasible for large-scale datasets, while the attribute importance-based has low time complexity. Moreover, tolerance relation is the weakened form of indistinguishable relations, which can validly solve incomplete information. Maximal consistent block describes the maximal objects set under the tolerance relationship, meaning that there is neither redundant, irrelevant information nor information loss. In contrast, the maximal consistent block accurately expresses the objects’ information under coverage and has higher accuracy.

After weighing the applicability of these variants, this paper introduces a maximal consistent block to deal with the uncertainty, incompleteness, and redundancy of data in the telecom fraud detection problem for the first time. Guided and inspired by previous research, an anomaly detection method (MCIR-CGAD) based on correlation and the maximal consistent block is proposed in this paper. The main contributions of this paper are summarized as follows:From the perspective of improving data quality based on the entropy function under rough set theory, we analyze the effect of attribute correlation and independence on the importance of attributes. A max-correlation and max-independence rate attribute reduction algorithm(MCIR) is designed to eliminate redundancy and noise contained in the data.From the perspective of data incompleteness processing, a rough gain anomaly detection algorithm (RGAD) is constructed based on the maximal consistent blocks and information gain, which can effectively supply missing data and provide an effective solution for incomplete data processing and feature information measurement.The effectiveness of the MCIR-RGAD algorithm is verified in the UCI dataset and authentic telecom fraud dataset. The results show that compared with the other eight kernel functions, the MCIR-RGAD algorithm can reduce the time complexity and effectively use the information contained in the missing data to improve the model performance.

The remainder of this paper is organized as follows. Section 2 gives the basic preliminaries of rough set theory. The design of the MCIR-RGAD algorithm is proposed in Section 3. Furthermore, Section 4 conducts the experimental analysis, and Section 5 summarizes the conclusions.

## 2. Preliminaries

### 2.1. Rough Set Theory

Rough set theory is an effective way to tackle and utilize incomplete datasets. The information contained in datasets can be represented as an information system.

An information system (U,A,V,f) is a decision information system, where $U=\{x_{1},x_{2},\cdots ,x_{n}\}$ is a nonempty finite set of objects known as a universe. Set $A=C⋃D=\{a_{1},a_{2},\cdots ,a_{m},D\}$ is composed of the condition attribute set $C=\{a_{1},a_{2},\cdots ,a_{m}\}$ and the decision attribute set *D*, where C⋂D=∅. The information function f:U×A→V is a map from the attribute of an object to information value, i.e., f(U,A)=V. Normally, a decision information system (U,A,V,f) can be abbreviated as (U,A).

**Definition** **1**(Indistinguishable Relation [37]). *Given an information system (U,A), A=C⋃D, ∀B⊆C is an attribute subset. An equivalence relation on the set U is called the indistinguishable relation IND(B), if it satisfies:*
(1)IND(B)={(x,y)∈U×U∣∀a∈B,f(x,a)=f(y,a)},*where ${\left[x\right]}_{IND(B)}=\left\{y|\left(x,y\right)\;\in \;IND\left(B\right)\;\right\}$ is a set of equivalence relations about x. Set family $U/IND\left(B\right)=\left\{{\left[x\right]}_{IND(B)}|x\in U\right\}=\left\{X_{1},X_{2},\cdots ,X_{m}\right\}$ means a partition of U about attribute set B. $U=\cup_{i=1}^{m}X_{i}$ and $X_{i}\cap X_{j}=\phi \left(i\neq j\right)$. Normally, ${\left[x\right]}_{IND(B)}$ and U/IND(B) can be abbreviated as ${\left[x\right]}_{B}$ and U/B, respectively.*

In an incomplete information system, the indistinguishable relation is unable to effectively divide the incomplete information. Then, the tolerance relation is given as follows.

**Definition** **2**(Tolerance Relation [37]). *Given an incomplete information system (U,A), A=C⋃D. ∀B⊆C is an attribute subset. The binary relation of incomplete information on U is defined as*
(2)$$\;SIM\left(B\right)\;=\left\{\left(x,y\right)\in U\;\times \;U|f\left(x,a\right)\;=\;f\left(y,a\right),or\;\;f\left(x,a\right)=*,or\;\;f\left(y,a\right)=*,\forall a\in B\right\},$$
*where * means the incomplete information. Denote U/SIM(B) as the family of all equivalence classes of SIM(B), or simply U/B.*

**Definition** **3**(Maximal Consistent Block [31]). *Given an incomplete information system (U,A), A=C⋃D, B⊆C is an attribute subset, and Y is said to be a maximal consistent block of attribute set B. If Y satisfies*
(i)*∀x,y∈Y⊆U, s.t. (x,y)∈SIM(B),then Y is called a consistent block;*(ii)*∄X∈MCB(B), s.t. Y⊆X.**where MCB(B) is the set of all maximal consistent blocks with B⊆A, ∀x∈U. The set of all MCB of x is denoted by $MCB_{x}\left(B\right)$, where $MCB_{x}\left(B\right)=\left\{Y|Y\in MCB\left(B\right),x\in Y\right\}$.*

**Example 1**.
*Consider descriptions of several users of the telecom network in Table 1. It is an incomplete decision information system (U,A), A=C⋃D, where $U=\{x_{1},x_{2},\cdots ,x_{5}\}$, $A=\{a_{1},a_{2},a_{3}\}$ with $a_{1}$-Duration, $a_{2}$-Place, $a_{3}$-Platform, and * means the incomplete information.*


According to the tolerance relation in Definition 2, it follows that $U/A=\{{\left[x_{1}\right]}_{A},\cdots ,$${\left[x_{5}\right]}_{A}\}$, where ${\left[x_{1}\right]}_{A}={\left[x_{5}\right]}_{A}=\left\{x_{1},x_{5}\right\}$, ${\left[x_{2}\right]}_{A}=\left\{x_{2},x_{3}\right\}$, ${\left[x_{3}\right]}_{A}=\left\{x_{2},x_{3},x_{4}\right\}$, ${\left[x_{4}\right]}_{A}=\left\{x_{3},x_{4}\right\}$.

By the concept of the maximal consistent block in Definition 3, the maximal consistent block of attribute set A is $MCB\left(A\right)=\{{\left[x_{1}\right]}_{A},{\left[x_{2}\right]}_{A},{\left[x_{4}\right]}_{A}\}$.

**Definition** **4**(Information Granularity [37]). *Given an incomplete information system (U,A), A=C⋃D, ∀B⊆C is an attribute subset, and the information granularity of attribute B is defined as*
(3)$$G\left(B\right)=\frac{1}{|U|}\sum_{i=1}^{|U|}\frac{|{\left[x_{i}\right]}_{B}|}{|U|},$$
*where $\;|{\left[x_{i}\right]}_{B}|\;$ and ∣U∣ mean the number of the indistinguishable relation set $\;{\left[x\right]}_{B}\;$ and set U, respectively.*

**Remark 1**.
*Given an incomplete information system (U,A), A=C⋃D, ∀B⊆C, conditional granularity, mutual information granularity, and joint granularity of attribute set B and D are defined as [28,38] $CG\left(B|D\right)=\sum_{i=1}^{|U|}\frac{|{\left[x_{i}\right]}_{B}|-|{\left[x_{i}\right]}_{B}\cap {\left[x_{i}\right]}_{D}|}{|U{|}^{2}}$, $MG\left(B;D\right)=\sum_{i=1}^{|U|}\frac{\;|{\left[x_{i}\right]}_{B}|\;+\;|{\left[x_{i}\right]}_{D}|\;-\;|{\left[x_{i}\right]}_{B}\cap {\left[x_{i}\right]}_{D}|\;}{|U{|}^{2}}$, $JG\left(B\cup D\right)=\sum_{i=1}^{|U|}\frac{|{\left[x_{i}\right]}_{B}\cap {\left[x_{i}\right]}_{D}|}{|U{|}^{2}}$, where $|{\left[x_{i}\right]}_{B}\cap {\left[x_{i}\right]}_{D}|=|{\left[x_{i}\right]}_{B⋃D}|$ means the division of knowledge under attribute B and attribute D.*


### 2.2. Information Theory

Information entropy is a measure of system uncertainty from the perspective of an information view. The magnitude of entropy reflects the degree of chaos or uncertainty of the system through the distribution of data information.

**Definition** **5**(Information Entropy [37]). *Given an incomplete information system (U,A), A=C⋃D, ∀B⊆C is an attribute subset, and the information entropy H(B) is defined as*
(4)$$H\left(B\right)=-\frac{1}{|U|}\sum_{i=1}^{|U|}\log_{2}\left(\frac{|{\left[x_{i}\right]}_{B}|}{|U|}\right),$$
*where ∣U∣ means the element number of object set U.*

**Remark 2**.
*By [37], the information entropy is called the granulation measure. The equivalent definition of the complete information system (U,A) in Equation (Equation 4) is defined as*

(5)
$$H\left(B\right)=-\sum_{i=1}^{|U/B|}\frac{|X_{i}|}{|U|}\log_{2}\frac{|X_{i}|}{|U|},$$

*where $U/B=\left\{{\left[x\right]}_{B}|x\in U\right\}=\left\{X_{1},X_{2},\cdots ,X_{m}\right\}$, ∣U/B∣=m, and $X_{i}⋂X_{j}=\phi$, i≠j. The form of Equation () is consistent with the basic definition of information entropy $H\left(X\right)=-\sum_{i=1}^{|U/B|}p_{i}\log_{2}p_{i}$, where $\;p_{i}\;=\;\frac{|X_{i}|}{|U|}$, $\;\sum_{i=1}^{m}\;p_{i}\;=\;1$. Therefore, we can understand the change of information entropy from the relationship between sets by a Venn diagram. In the information theory of rough set theory, the finer the partition, the bigger the entropy.*


**Remark 3**.
*In a complete information system (U,A), condition entropy H(D∣B), mutual information H(D;B), and joint entropy H(D∪B) of attribute B and D are defined as [39] $H\left(D|B\right)=-\sum_{i=1}^{m}\sum_{j=1}^{n}\frac{|X_{i}\cap Y_{j}|}{|U|}\log_{2}\frac{|X_{i}\cap Y_{j}|}{|X_{i}|}$, $H\left(D;B\right)=-\sum_{i=1}^{m}\sum_{j=1}^{n}\frac{|X_{i}\cap Y_{j}|}{|U|}\log_{2}\frac{|X_{i}|\cdot |Y_{j}|}{|X_{i}\cap Y_{j}|\cdot |U|}$, $H\left(D\cup B\right)=-\sum_{i=1}^{m}\sum_{j=1}^{n}\frac{|X_{i}\cap Y_{j}|}{|U|}\log_{2}\frac{|X_{i}\cap Y_{j}|}{|U|}$.*


**Theorem** **1**(Entropy Measure). *Given an incomplete information system (U,A), A=C⋃D, ∀B⊆C, conditional entropy, mutual information, and joint entropy of attribute B and D are defined as*
(6)$$\begin{array}{cc} (i)CH(D|B)\; & \;=\;-\frac{1}{|U|}\sum_{i=1}^{|U|}\log_{2}\frac{|{\left[x_{i}\right]}_{B}\cap {\left[x_{i}\right]}_{D}|}{|{\left[x_{i}\right]}_{B}|},\\ (ii)MH(D;B)\; & \;=\;-\frac{1}{|U|}\sum_{i=1}^{|U|}\log_{2}\frac{\;|{\left[x_{i}\right]}_{B}|\;\cdot \;|{\left[x_{i}\right]}_{D}|\;}{\;|{\left[x_{i}\right]}_{B}\cap {\left[x_{i}\right]}_{D}|\;\cdot \;|U|\;},\\ (iii)JH(D\cup B)\; & \;=\;-\frac{1}{|U|}\sum_{i=1}^{|U|}\log_{2}\frac{|{\left[x_{i}\right]}_{B}\cap {\left[x_{i}\right]}_{D}|}{|U|}, \end{array}$$
*where MH(D;B)=H(D)−CH(D∣B).*

**Proof of Theorem 1** The specific proof of Theorem 1 can be found in Appendix A.    □

## 3. Max-Correlation and Max-Independence Rate-Rough GAIN anomaly Detection Algorithm (MCIR-RGAD)

In the information age of Industry 4.0, the amount of data containing a large number of attributes has proliferated. However, not all attributes are relevant to the classification task. In cyberspace, data may be relevant, repetitive, or similar, which does not bring new and valuable information to the anomaly detection task, leading to unnecessary time costs. In addition, attributes that are not relevant to the anomaly detection task may be noisy and not only fail to help model learning, but may even affect detection performance. In addition, data may inevitably be lost during collection, processing, and storage. The lost data itself may contain hidden anomaly information, and a simple subjective assignment or deletion may lead to an invalid use of the lost information. As a result, the attributes in the data are usually not fully functional.

The entropy function of information theory, as a quantitative paradigm for measuring uncertainty, can effectively measure the correlation between attributes in data. In addition, the missing incomplete data has a certain degree of uncertainty, and this uncertainty may also contain valuable information. Therefore, this paper uses the mutual information function in information theory to measure important attributes that are highly relevant and less redundant to the classification task. Further, the maximal consistent blocks of rough set theory are used to process the missing data, and the information useful for the anomaly detection task is mined from the perspective of uncertainty of incomplete data to realize the improvement of the anomaly detection performance.

The main idea of this section can be divided into three parts. Section 3.1 theoretically discusses the relationship between the attribute correlation and redundancy in the incomplete information system. Then, Section 3.2 presents an optimization algorithm of attribute reduction(MCIR) with a correlation and independent information. Further, we design a rough gain anomaly detection algorithm(RGAD) based on the maximal consistent block to solve the incompleteness of authentic telecom fraud detection in Section 3.3. Figure 1 shows the framework of the proposed methodology.

### 3.1. Relationship of Correlation and Redundancy

To date, many criteria have been proposed to consider the correlation or redundancy of new classification information, such as criteria JMI, CMIM, CIFE, ICAP, ICI, MCI, etc., which are summarized in [19]. The criteria are shown as follows.

$J_{JMI}\left(a_{j}\right)=\sum_{a_{i}\in B}MH\left(a_{i},a_{j};D\right)≜\sum_{i}$(①${}_{i}$ + ②${}_{i}$ + ③${}_{i}$),

$J_{CIFE}\left(a_{j}\right)=MH\left(D;a_{j}\right)-\sum_{a_{i}\in B}MH\left(D;a_{j};a_{i}\right)≜$ ②${}_{i}+$③${}_{i}-\sum_{i}$②${}_{i}$,

$J_{ICAP}\left(a_{j}\right)=MH\left(D;a_{j}\right)-\sum_{a_{i}\in B}\max MH\left(D;a_{j};a_{i}\right)≜$ ②${}_{i}$ + ③${}_{i}-|B|\max_{i}$②${}_{i}$,

$J_{ICI}\left(a_{j}\right)=MH\left(D;a_{i}|a_{j}\right)+MH\left(D;a_{j}|a_{i}\right)≜$ ①${}_{i}$ + ③${}_{i}$,

$J_{MRI}\left(a_{j}\right)=MH\left(D,a_{j}\right)+\sum_{a_{i}\in B}ICI\left(D;a_{i},a_{j}\right)≜$②${}_{i}$ + ③${}_{i}$ + $\sum_{i}$(①${}_{i}$ + ③${}_{i}$),

where $MH\left(\;D;\;a_{i}\;|\;a_{j}\right)\;=\;MH\left(\;D;\;a_{i}\;,\;a_{j}\;\right)\;-\;MH\left(\;D;\;a_{j}\;\right)$, ①${}_{i}\;=\;MH\left(\;D;\;a_{i}\;|\;a_{j}\;\right)$, ②${}_{i}\;=\;MH\left(\;D;\;a_{i};\;a_{j}\;\right)$, and ③${}_{i}\;=\;MH\left(\;D;\;a_{j}\;|\;a_{i}\;\right)$.

In Figure 2a, ①${}_{i}$ manifests the relevant information of selected attribute $a_{i}\in B$, ②${}_{i}$ means the redundant information between the attributes $a_{i}$, $a_{j}$, and D, and ③${}_{i}$ represents the relevant information of candidate attribute.

In the literature, the correlation and redundancy of the criterion function are frequently compared between each candidate attribute $a_{j}$ and each attribute $a_{i}$ of the selected attribute set B. However, this comparison method has a lot of redundant calculations about information. Therefore, this paper regards the selected attribute reduction set B as a whole and studies the correlation and redundancy between the candidate attribute $a_{j}$ and the attribute reduction set B, as shown in Figure 2b.

For the convenience of formulation, we set ①, ②, and ③ denoted as ① $=\;MH(D;\;B\;|\;a_{j}\;)$, ② $\;=\;MH(\;D;\;B;\;a_{j}\;)$, and ③ $=MH(D;a_{j}|B)$, respectively. In Figure 2b, ① manifests the relevant information of selected attribute $a_{i}\in B$, ② means the redundant information between the attributes B, $a_{j}$, and D, and ③ represents the relevant and independent information of candidate attribute $a_{j}$.

**Theorem** **2**(③ ≜ ① + ② + ③). *Given an incomplete information system (U,A), A=C⋃D, B⊆C has been selected, and $a_{j}$ is a candidate attribute, then the correlation and redundancy relationship between the attribute $a_{j}$, B, and D satisfies*
(7)$$MH\left(D;a_{j}|B\right)≜MH\left(D;B\cup a_{j}\right).$$

**Proof of Theorem 2** According to the definition of symbols ①, ②, and ③, we deduce$MH\left(D;B\cup a_{j}\right)-MH\left(D;a_{j}|B\right)=$ (① +② + ③) − ③ =MH(D;B).In the decision information system, the attribute set B has been selected, and D is certain, so the division of knowledge is definite. Then, MH(D;B) is a constant. There is a nonnegative constant $\Delta_{3}=MH\left(D;B\right)$, such that
(8)$$MH\left(D;B\cup a_{j}\right)-MH\left(D;a_{j}|B\right)=\Delta_{3}.$$Hence, Equation (Equation 7) holds, i.e., ③ ≜ ① + ② + ③.    □

Theorem 2 manifests that the correlation of the newly selected attribute $a_{j}$ is consistent with the attribute reduction set $B\cup a_{j}$, and the effect is the same in classification detection.

**Theorem** **3**(① + ③ ≜ ③ − ②). *Given an incomplete information system (U,A), A=C⋃D, suppose attribute set B⊆C has been selected, and $a_{j}$ is a candidate attribute, then the correlation and redundancy relationship between the attribute $a_{j}$, B, and D satisfies*
(9)$$\begin{array}{c} MH\left(D;B|a_{j}\right)+MH\left(D;a_{j}|B\right)≜MH\left(D;a_{j}|B\right)-MH\left(D;B;a_{j}\right). \end{array}$$

**Proof of Theorem 3** According to the definition of symbols ①, ②, and ③, we have that $|\left(MH\left(D;B|a_{j}\right)+MH\left(D;a_{j}|B\right)\right)-\left(MH\left(D;a_{j}|B\right)\;-\;MH\left(D;B;a_{j}\right)\right)|=|$(① + ③) − (③ − ②)∣.Based on Equation (Equation 8) in Theorem 2, ③ $+\Delta_{3}=$ ① + ② + ③ holds, hence
(10)$$|\left(MH\left(D;B|a_{j}\right)+MH\left(D;a_{j}|B\right)\right)-\left(MH\left(D;a_{j}|B\right)\;-\;MH\left(D;B;a_{j}\right)\right)|$$=∣(① + ③) − (③ − ②)∣=∣(① + ③) − [(① + ② + ③ −$\Delta_{3}$) − ②]∣=∣(① + ③) − (① + ③ −$\Delta_{3}$)∣ = $\Delta_{3}$Hence, Theorem 3 is proved, i.e., ① + ③ ≜ ③ − ②.    □

Theorem 3 shows that only the correlation between the new attribute $a_{j}$ and the selected attribute set B is considered, which is equivalent to considering the correlation and redundancy of new attributes $a_{j}$.

### 3.2. Max-Correlation and Max-Independence Rate Algorithm (MCIR)

In light of the above analysis and inspired by the literature [19], the max-correlation and max-independence rate algorithm (MCIR) is introduced as follows.

**Definition** **6**(MCIR). *Given an incomplete information system (U,A), A=C⋃D, B⊆C, suppose $a_{i}\in B$ and $a_{j}\in C-B$, then the max-correlation and max-independence rate function is presented as*
(11)$$a_{j}=\underset{a_{j}\in C-B}{\arg}\max J_{MCIR}\left(a_{j}\right),$$
*where $J_{MCIR}\left(a_{j}\right)=\left\{\frac{MH(D;a_{j}|B)}{H(a_{j})}|\min_{a_{j}\in C-B}MH\left(D;B;a_{j}\right)\right\}$, i.e.$J_{MCIR}\left(a_{j}\right)=\{$③$/H(a_{j})|\min$②}.*

The principle of the MCIR algorithm is to maximize the correlation and the independence of new classification information and minimize the redundancy between old attributes. The definition of information entropy in rough set theory is from the view of the object attribute information division. The finer the division, the greater the entropy value. Therefore, when the system increases the correlation, it tends to select attributes with more new information.

The attribute reduction algorithm based on max-correlation and max-independence rate is shown in Algorithm 1.
**Algorithm 1:**Max-Correlation and Max-Independence Rate (MCIR)**Input:** Information system (U,C⋃D).**Output:** An attribute reduction set B.1:compute MH(C;D)2:Feat⇐C,3:$B⇐\underset{a_{j}\in Feat}{\arg}\max\;|\;MH\left(D;B\right)\;-\;MH\left(D;B⋃a_{j}\right)\;|$4:Feat⇐C−B,5:**while**$J_{MH}\left(a_{j}\right)\geq \theta$**do**6:   **for** $a_{j}\in Feat$ **do**7:     **if** ∣Feat∣=0 **then**8:        B⇐B9:     **else**[∣Feat∣≠0]10:        $a_{j}⇐\underset{a_{j}\in Feat}{\arg}\max J_{MCIR}\left(a_{j}\right)$11:        $B⇐B+\{a_{j}\}$12:        $Feat⇐Feat-\{a_{j}\}$13:        $\;J_{MH}\left(a_{j}\right)\;⇐\;|\;CH\left(D\;|\;C\right)\;-\;CH\left(D\;|\;B\right)\;|\;$14:     **end if**15:   **end for**16:**end while**

With the data obtained in the different scenarios, the importance of the correlation and redundancy between attributes exists in diversity. In other words, in the incomplete information system, when the effect of correlation is far biggerer than the redundancy, it is more effective to add new information related to the decision attribute. When similar, redundant, and repetitive information causes noise to affect the detection and classification, it is necessary to increase the correlation and reduce the redundancy.

From the relationship of relevance and independence of the MCIR algorithm in Definition 6, Figure 2c satisfies ③${}_{j}<$ ③${}_{k}=$③${}_{l}$, ②${}_{l}=$ ②${}_{j}$. It shows that the order of attribute importance is $a_{k}⪰a_{l}⪰a_{j}$, i.e., attribute $a_{k}$ is better than attribute $a_{i}$, and attribute $a_{i}$ is better than attribute $a_{j}$, which can be sorted correctly by the MCIR algorithm.

### 3.3. Rough Gain Anomaly Detection Algorithm with Max-Correlation and Max-Independence Rate (MCIR-RGAD)

An anomaly detection algorithm (MCIR-RGAD) is designed based on the maximal consistent block horizontally supplementing reduced data. Then, anomaly detection is carried out for the new complemented data. Inspired by the design of information gain in the decision tree, the main idea of the MCIR-RGAD algorithm is to construct a correlation function to measure the ability of attribute classification.

The decision tree, one of the basic classification methods of machine learning, achieves classification tasks by the characteristics of data information. It has fast classification speed, strong interpretability, and readability. Generally, the decision tree learning process consists of feature selection, decision tree generation, and decision tree pruning. In the decision tree, to improve the learning efficiency of the decision tree, the kernel functions, such as information gain, information gain rate, or Gini coefficient, are used to select important features, and then, the decision tree is constructed recursively based on the kernel function. To avoid the occurrence of classification overfitting, we prune the decision tree to balance the model complexity while ensuring the fitting accuracy of the training data.

Both attribute reduction and decision tree work by finding significant features that can classify decision features in information systems. Attribute reduction algorithms can effectively find relevant classification features and achieve effective feature selection. In addition, since there may be intersections in the equivalence class of the object set divided by the maximally consistent block in the incomplete information system, the completeness is not satisfied, i.e., $\sum_{i}p_{i}\geq 1$, and there is a negative value when using the information gain for decision learning. Therefore, this paper designs an improved algorithm(MCIR-RGAD) to solve the anomaly detection problem in incomplete systems. Moreover, similar, redundant, repeated, or invalid features are filtered out by reducing. Therefore, this paper does not consider decision pruning.

Frequently, missing data is handled simply by deleting the missing row, filling in zero, filling in one, or filling in the previous data information. However, the explicit deletion or subjective filling of the acquired information will destroy the original data information, so that the missing information cannot be effectively utilized and processed. In an incomplete information system, knowledge can be divided according to the compatibility between available and missing information. This division method not only does not lose the existing data information, but also is more objective. The definition of the kernel function, rough gain RG, is given below.

**Definition** **7**([40] Rough Entropy). *Given an incomplete information system (U,A), A=C⋃D, ∀B⊆C, rough entropy $E_{r}\left(B\right)$ is defined as*
(12)$$E_{r}\left(B\right)=-\sum_{i=1}^{U}\frac{1}{|U|}\log_{2}\frac{1}{|{\left[x\right]}_{B}|},$$
*where rough entropy $E_{r}\left(B\right)$ satisfies $E_{r}\left(B\right)+H\left(B\right)=\log_{2}|U|$.*

Inspired by the literature [40], this paper presents a generalized form of the definition of rough entropy for decision making in information division as shown in Definition 8.

**Definition** **8**(Decision Rough Entropy). *Given an incomplete information system (U,A), A=C⋃D, ∀B⊆C, the maximal consistent block of attributes B and D are $MCB\left(B\right)=\left\{B_{1},\cdots ,B_{k}\right\}$, $MCB\left(D\right)=\left\{D_{1},\cdots ,D_{m}\right\}$, then the decision rough entropy $E_{r}\left(D_{B}\right)$ is defined as*
(13)$$E_{r}\left(D_{B}\right)=-\sum_{j=1}^{m}\sum_{i=1}^{k}\frac{|B_{i}|}{|U|}\log_{2}\frac{|B_{i}⋂D_{j}|}{|B_{i}|}.$$

**Definition** **9**(Rough Gain). *Given an incomplete information system (U,A), A=C⋃D, ∀B⊆C, the maximal consistent block of attributes B and D are $MCB\left(B\right)=\left\{B_{1},\cdots ,B_{k}\right\}$, $MCB\left(D\right)=\left\{D_{1},\cdots ,D_{m}\right\}$, then the rough gain are defined as*
(14)$$RG\left(D,B\right)=g\cdot R_{r}\left(D,B\right)+\left(1-g\right)\cdot \frac{1}{G_{r}\left(D,B\right)},$$
*where g∈[0,1] is a positive constant, $R_{r}\left(D,B\right)=\frac{E_{r}\left(D_{B}\right)}{E_{r}\left(B\right)}$ is the rough entropy rate, $E_{r}\left(D_{B}\right)$ is decision rough entropy, $E_{r}\left(B\right)$ is rough entropy of attribute B, $G_{r}\left(D,B\right)=\frac{G(D,B)}{H(B)}$ is the information gain rate, G(D,B)=H(D)−H(D∣B) is the information gain, $H\left(D\right)=-\sum_{i=1}^{|m|}\frac{|D_{i}|}{|U|}\log_{2}\frac{|D_{i}|}{|U|}$, and $H\left(D|B\right)=-\sum_{i=1}^{k}\sum_{j=1}^{m}\frac{|B_{i}\cap D_{j}|}{|U|}\log_{2}\frac{|B_{i}\cap D_{j}|}{|B_{i}|}$.*

Therefore, this paper selects features based on the MCIR algorithm, then combines the advantage of the information gain with rough entropy to deal with missing data information. We design an anomaly detection algorithm, MCIR-RGAD algorithm, to achieve the task of anomaly detection. The specific algorithm is shown in Algorithm 2.

Essentially, the MCIR-RGAD algorithm replaces the information gain function of the decision tree with the rough gain function in Definition 9. Contrary to the information gain, a smaller rough gain indicates a better attribute, and the other parts are consistent with the decision tree. Therefore, consistent with the decision tree model, the time complexity of this model is $O(n^{2})$.
**Algorithm 2:**MCIR-RGAD algorithm**Input:** Information system (U,C⋃D), an attribute reduction set B, threshold ϵ>0.**Output:** A decision tree *T*1:compute B   ▹UsingAlgorithm2:compute $MCB\left(B\right)=\left\{B_{1},\cdots ,B_{k}\right\}$, $MCB\left(D\right)=\left\{D_{1},\cdots ,D_{m}\right\}$.  ▹UsingDefinition3:**if**(U,C⋃D) is incomplete **then**4:   g⇐1, $RG\left(D,B\right)⇐R_{r}\left(D,B\right)$ (Equation (Equation 14))5:**else**6:   0<g<1, $RG\left(D,B\right)⇐g\cdot R_{r}\left(D,B\right)+\left(1-g\right)\cdot \frac{1}{G_{r}\left(D,B\right)}$ (Equation (Equation 14))7:**end if**▹RecursionPoint8:**for**$a_{j}\in B$**do**9:   $\;a_{j}\;⇐\;\underset{a_{j}\in B}{\arg}\;\min\;RG\left(D,a_{j}\right)$10:   $B\;⇐\;B\;-\;a_{j}\;$11:   **if** $RG(D,a_{j})>\epsilon$ **then**12:     $Label_{T}⇐\underset{f(U,D_{i})}{\arg}\;\max\;|D_{i}|\;$13:   **else**14:     $MCB\left(a_{j}\right)⇐\left\{X_{1},\cdots ,X_{s}\right\}$15:     **for** $X_{i}\in MCB\left(a_{j}\right)$ **do**16:        $\left(U,C⋃D\right)⇐\left(X_{i},B⋃D\right)$17:        **if** ∣MCB(D)∣=1 **then**18:          $Label_{T}⇐f\left(U,D\right)$ B=ϕ19:          $Label_{T}⇐\underset{f(U,D_{i})}{\arg}\;\max\;|D_{i}|$20:        **else**21:          return Step 822:        **end if**23:     **end for**24:   **end if**25:**end for**26:return *T*

## 4. Experimental Analysis

The UCI Machine Learning Repository datasets (https://archive.ics.uci.edu/ml/index.php accessed on 12 April 2022) and the Sichuan telecom fraud phone datasets (https://aistudio.baidu.com/aistudio/datasetdetail/40690 accessed on 12 April 2022) are used to verify the effectiveness of the proposed method in this section. The MCIR-RGAD the orithms are coded in Python using Visual Studio Code and were run on a remote server with a GPU, NVIDIA GeForce RTX 3090, 48 RAM.

The Sichuan telecom fraud phone dataset consists of four datasets, namely call data (VOC), short message service data (SMS), user information data (USER), and Internet behavior data (APP). The Union data is an integrated dataset combined based on user phone numbers and contains the attribute of four datasets: Voc dataset, APP dataset, SMS dataset, and User dataset. The details of the datasets are described in Table 2.

The goal of this paper is to detect fraudulent users among regular users according to the important attribute efficiently selected by the correlation and independence from the perspective of data uncertainty and incompleteness. Next, we discuss the effectiveness of the method proposed in this paper from three aspects: incompleteness of data (Definition 3), MCIR attribute reduction algorithm (Section 3.2), and MCIR-RGAD anomaly detection classifier (Section 3.3).

### 4.1. Incompleteness of Data

Loss of data during recording, storage, or transmission is a very likely problem. Normally, the way to deal with incomplete information is to delete it directly, fill it with zeero, one or mean value; however, this simple way of dealing with it will cause the loss of information. As can be seen from Figure 3, in the Sichuan Telecom fraud dataset, most of the users with null values (red parts) are abnormal, and if they are directly deleted or simply assigned, the abnormal information will not be effectively used. Therefore, from the perspective of improving data quality, this paper uses the idea of maximal consistent blocks in rough set theory to deal with incomplete data to achieve effective information mining.

Then, Table 3 and Figure 4 further illustrate the effectiveness of the maximal consistent block in handling incomplete data. Table 3 and Figure 4 are the performance comparisons of tackling null values under authentic incomplete telecom fraud data and random deletion of artificially constructed incomplete data (5%, 10%,*…*, 50%, 10 types of data missing ratios). From the perspective of the accuracy (Figure 4a), recall (Figure 4b), F1 (Figure 4c), and the number of correct predictions, the maximal consistent block (MCB) can effectively utilize incomplete information and avoid unnecessary information loss.

### 4.2. Attribute Reduction under MCIR Algorithm

This paper proposes an attribute reduction algorithm of MCIR which uses the entropy function to measure the correlation and independence of attributes from the perspective of rough set theory. The calculation time of the algorithm is reduced while ensuring the accuracy of the telecom fraud detection problem. The main idea is to reduce the computation time by filtering out partial attributes that are most relevant to fraudulent users and have the greatest independence (least redundancy).

Experiments on UCI and telecom fraud data show that the computation time of the data can be significantly reduced by filtering out important attributes. Figure 5 and Figure 6 further illustrate that the MCIR algorithm not only effectively reduces the computation time, but also eliminates the adverse effects of noise on information, improves data quality, and maintains or even improves the accuracy of model detection.

Generally, datasets can be roughly divided into four types, namely: non-redundant and noise-free dataset (Figure 7a Car, approximated as a strictly monotonically increasing function), non-redundant and noisy dataset (Figure 7b Adult, approximately concave function), redundant and noisy dataset (Figure 7c Bank, approximately non-increasing function), and redundant and non-noise dataset (Figure 7d Mushroom, approximately non-decreasing function). Redundancy shows the approximation, repetition, and correlation of attributes in the data with each other; noise refers to the interference and misleading effects of certain attributes in the data on the classification task. Specifically, for a non-redundant and noise-free dataset, there is no need to perform attribute reduction, and each dimension of features is important information. For other types of data, it is necessary to remove redundant and noisy attributes. In addition, it can be seen from Figure 7 that compared with other different attribute reduction algorithms ($J_{CG}$ [41], $J_{CH}$[42], $J_{MG}$, $J_{MH}$ [43], $J_{JG}$, $J_{JH}$, ③, ① + ② + ③ [44], ① + ③ [19], ③–② [45]), the MCIR algorithm (red dotted line) designed in this paper achieves better accuracy with fewer attributes. Since the MCIR model removes as many redundant or noisy attributes from the data as possible and achieves data optimization through data dimensionality reduction, making the reduced data better for anomaly detection tasks, the model can maintain or even improve the accuracy of performance detection while reducing the time complexity.

Therefore, in the process of data processing, the MCIR algorithm can use partial important attribute information to shorten the computation time and effectively improve the detection accuracy of the model (Figure 7, the black dotted line).

Next, the feature selection of the telecom fraud dataset under the MCIR algorithm is discussed. Figure 8 shows correlations within attributes via a heatmap. Among them, Figure 8a,b are the correlations before and after attribute reduction, respectively. As shown in Figure 8, when attribute reduction is not performed, the data contain a lot of redundant information (dark patches). This paper constructs the MCIR attribute reduction algorithm from the perspective of attribute uncertainty and correlation, which can reduce the information redundancy degree of data while reducing data weight.

Further, the boxplot and probability distribution plot in Figure 9 show the difference in statistical distribution between normal and abnormal users. The important attributes selected based on the MCIR-RGAD algorithm can effectively highlight the difference between abnormal users and normal users, and fraudulent users can be filtered out by the selected important attributes. Compared with the original Union dataset of 84 attributes with 85.84% detection performance (Table 4), the detection performance of 10 attributes after MCIR simplification is improved to 89.96%, indicating that the MCIR model involved in this paper effectively achieves the selection of important attributes. To further visualize how the selected attributes distinguish between normal and fraudulent users, Figure 9 depicts the box line plot and statistical distribution of the 10 important attributes in the telecom fraud dataset filtered by the MCIR method. From Figure 9, it can be seen that the distributions of normal users and fraudulent users under the 10 attributes have large differences, mainly in the form of (a, f, e) with large difference in mean and variance, (b, d, j, g, h) with large difference in variances with similar means, and (c, i) with large difference in means with similar variances. The larger the difference between the mean and variance distributions of normal and fraudulent users for the selected attributes, the more effective it the method is in distinguishing fraudulent users.

### 4.3. Anomaly Detection under MCIR-RGAD Algorithm

Redundancy and noise attributes are removed from the original data to improve the data quality of the MCIR algorithm. Then, to perform effective anomaly detection on incomplete data containing missing content, this paper designs the MCIR-RGAD algorithm based on maximal consistent blocks. It provides an effective solution for the processing and utilization of incomplete data.

In the anomaly detection of the decision tree, six types of kernel function classification algorithms, namely Information Gain G(D,B), Information Gain Rate $G_{r}\left(D,B\right)$, Gini Coefficient, Rough Entropy $R_{r}\left(D,B\right)$, Rough Entropy Rate $R_{r}\left(D,B\right)$, and Rough Gain RG(D,B), are compared in this paper. As shown in Figure 10, the rough gain anomaly detection algorithm (RGAD) integrates rough entropy and information gain as the kernel function has better performance.

The performance and computation time of nine types of attribute reduction algorithms are shown in Table 5 and Table 6. Compared with other algorithms, the MCIR-RGAD algorithm proposed in this paper can effectively achieve classification detection in a shorter time.

To effectively measure the trade-off between detection performance and computation time cost of an algorithm, this paper designs a robustness metric in Definition 10. In the robustness metric, since computation time and performance level have different importance in different application scenarios, a linear parameter *k* is designed to trade off the importance of time and performance. The telecom fraud problem in this paper pays more attention to the accuracy of the model; hence, the hyperparameter weight in the robustness metric is set as k=0.4.

**Definition** **10**(Performance Robustness).
(15)Robust=k·T+(1−k)·P,
*where $\;P\;=\;\frac{P_{a}}{P_{b}}\;$ is the degree of performance retention, $T\;=\;\frac{T_{b}-T_{a}}{T_{b}}$ is the degree of time optimization, $P_{b}$, $P_{a}$, $T_{b}$, and $T_{a}$ are the performance and time before and after attribute reduction, respectively, and k∈[0,1] is a weight parameter of time, which means the importance of time cost.*

Then, Table 4 shows the number of attributes after attribute reduction for different datasets and shows the changes in performance and computation time of the MCIR-RGAD algorithm before and after attribute reduction. Note that this paper compares the performance and computation time of different algorithms in the same number of attribute reduction sets B.

The performance robustness metric with less computation time and high performance indicates that the designed classifier algorithm is better. The accuracies and computation time in Table 4 and Table 5 and Figure 11 show the robustness under the different attribute reduction algorithms. Compared with other algorithms, the MCIR-RGAD algorithm has strong robustness. That is, when the number of attributes in the attribute set is reduced to the same number, the anomaly detection algorithm MCIR-RGAD can effectively ensure the accuracy of classification detection while shortening the calculation time.

### 4.4. Statistical Test Analysis

Two nonparametric statistical test analyses of the Friedman test and the Nemenyi post hoc test are introduced to further verify the validity of the comparison method and the proposed method. We compare the performance differences at a significance level of α=0.05.

#### 4.4.1. Friedman Test

The Friedman test can effectively determine whether there is a significant difference in algorithm performance. Suppose we compare *K* algorithms on *N* datasets. In the Friedman test, the null hypothesis assumes that there is no significant difference between the models. First, the models were ranked on different datasets using the performance accuracy cases in Table 5. Then, we acquire the average of overall ranking for each model, $R_{ave}^{j}=\frac{1}{N}\sum_{i=1}^{K}r_{i}$. The performance ranking of the nine algorithms on the nine datasets is given in Table 7. When the performance of the algorithms is equal, the ordinal values are averaged. For example, if the performance of the 7 algorithms ($J_{CG}$, $J_{MG}$, $J_{CH}$, $J_{MH}$, ③, ①+③, $J_{MCIR-RGAD}$) under the Car dataset in Table 5 is equal, then their rank values are $r_{i}=\frac{1+2+3+4+5+6+7}{7}=4$.

The Friedman statistic $\tau_{\chi^{2}}=\frac{12N}{K\left(K+1\right)}\left(\sum_{i=1}^{K}r_{i}^{2}-\frac{K{\left(K+1\right)}^{2}}{4}\right)$ is distributed according to $\chi^{2}$-distribution with K−1 degrees of freedom, when *K* and *N* are large enough. Owing to the overly conservative nature of the original Friedman test, the variable $\tau_{F}=\frac{\left(N-1\right)\tau_{\chi^{2}}}{N\left(K-1\right)-\tau_{\chi^{2}}}$ is commonly used today, which is distributed according to F-distribution with K−1 and (K−1)(N−1) degrees of freedom, i.e., $\tau_{F}∼F\left(K-1,\left(K-1\right)\left(N-1\right)\right)$.

This paper compares nine algorithms using nine datasets. In the Friedman test, if the p-value is less than the significance level or the $\tau_{F}$ value is greater than the critical value F(8,64) determined by the F-distribution table, the null hypothesis can be rejected, and at least two algorithms are considered to have significant differences. By checking the table and calculating, we have $\tau_{F}=5.0208>F\left(8,64\right)=2.0868$ and $p=2.688\times 10^{-5}<0.05$. Therefore, the null hypothesis can be rejected with 95% confidence level, indicating that there is a significant difference between the algorithms in the model. Then, a pairwise comparison of the benchmark algorithms was performed using the Nemenyi post hoc test.

#### 4.4.2. Nemenyi Post Hoc Test

In the Nemenyi post hoc test, the performance of two models is considered to be significantly different if the average rank value $R_{ave}^{j}$ of the two models is greater than or equal to the criterion distance $\left(CD=q_{\alpha }\sqrt{\frac{K\left(K+1\right)}{6N}}\right)$, where the critical value $q_{\alpha }$ obeys the Tukey distribution. By checking the table and calculating, $q_{0.05}=3.102$ under the confidence level α=0.05, then CD=4.0047. It can be seen from Figure 12 that the MCIR-RGAD model is optimal and significantly different from $J_{JH}$ and $J_{JG}$. In addition, $J_{CH}$, $J_{MH}$, and 3 are equivalent, and $J_{MG}$ is equivalent to $J_{CG}$. Namely, the model performance can be ordered as $J_{MCIR-RGAD}>J_{CH}=J_{MH}=$③$>J_{MG}=J_{CG}>$①+③$>J_{JG}>J_{JH}$.

## 5. Conclusions

It is crucial and time-consuming to obtain anomaly classification information in big data with uncertainty, redundancy, and incompleteness. In this paper, a new attribute reduction algorithm (MCIR) is proposed based on the correlation and independence of the data. Furthermore, considering the consistency of attribute reduction and decision tree in selecting features, this paper combines their advantages and constructs an anomaly detection algorithm called RGAD to tackle incomplete data based on the maximal consistent blocks. The proposed algorithm (MCIR-RGAD) can significantly reduce the computation time and effectively maintain or improve the accuracy. Therefore, facing the problem of anomaly detection, this paper provides an effective solution for the optimization of data quality and the processing of incomplete data.

In the future, we plan to extend this work in the context of unsupervised learning from the perspective of structural information among objects, using the concept of neighborhood information systems in rough set theory. The extended work will optimize the data quality and reduce the time complexity through attribute reduction methods, improve the detection performance of classification tasks through structural information, and maximize valuable information through incomplete mixed data (both categorical and numerical data). This will provide an effective solution to the research of information theory and rough set theory on anomaly detection problems.

## Figures and Tables

**Figure 1 entropy-25-00112-f001:**
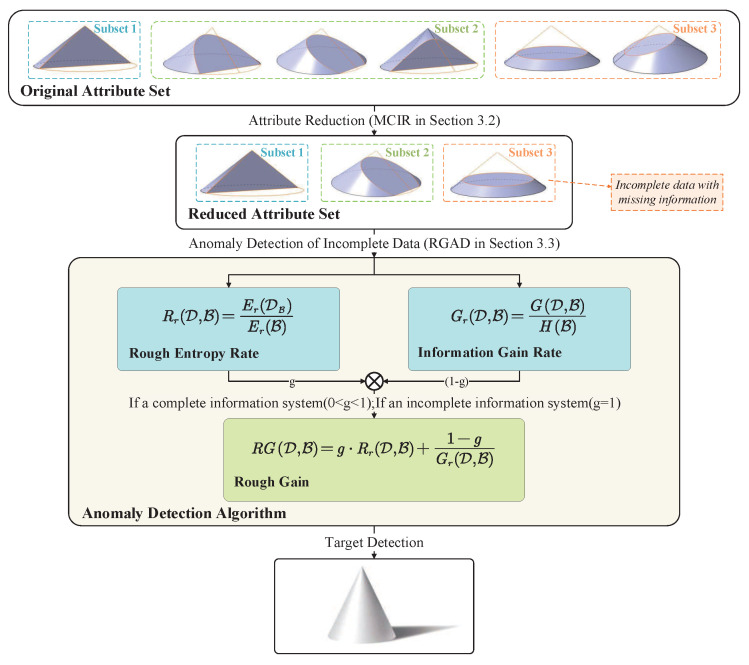
The framework of the proposed methodology.

**Figure 2 entropy-25-00112-f002:**
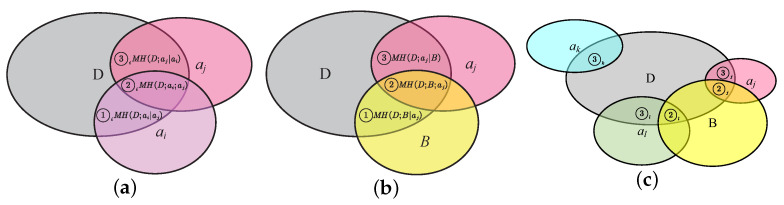
The relationship of relevance and independence in the complete information system: (**a**) $a_{i}$ is a variable, $a_{j}$ is fixed; (**b**) $a_{j}$ is a variable, *B* is fixed; (**c**) candidate attribute selection.

**Figure 3 entropy-25-00112-f003:**
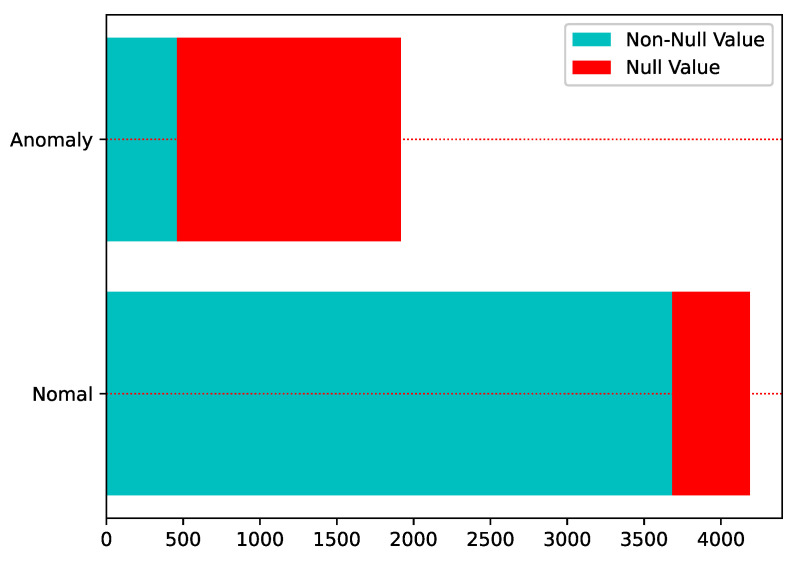
Incomplete data among telecom fraud users.

**Figure 4 entropy-25-00112-f004:**
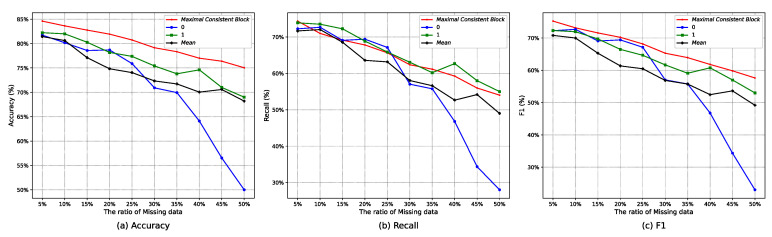
Performance comparison of randomly deleting missing data of the Union dataset under the RGAD algorithm.

**Figure 5 entropy-25-00112-f005:**
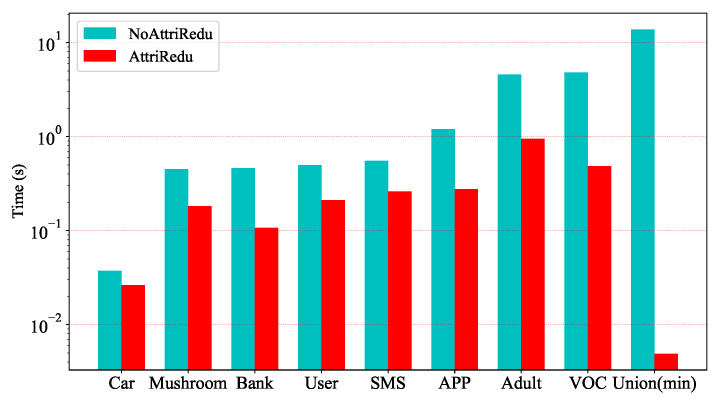
Comparison of the computation time before and after the MCIR-RGAD algorithm.

**Figure 6 entropy-25-00112-f006:**
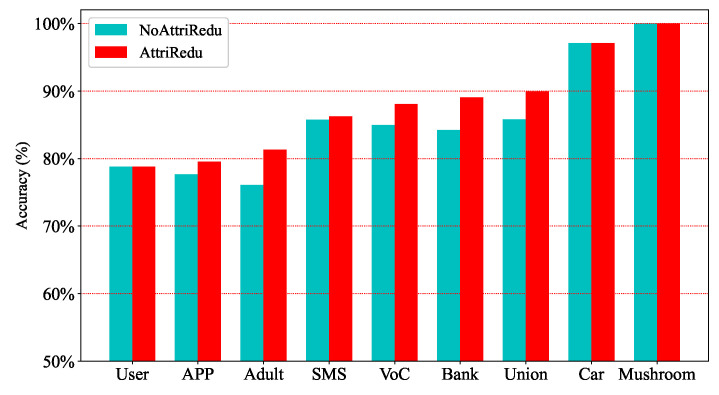
Comparison of the classification accuracy before and after the MCIR-RGAD algorithm.

**Figure 7 entropy-25-00112-f007:**
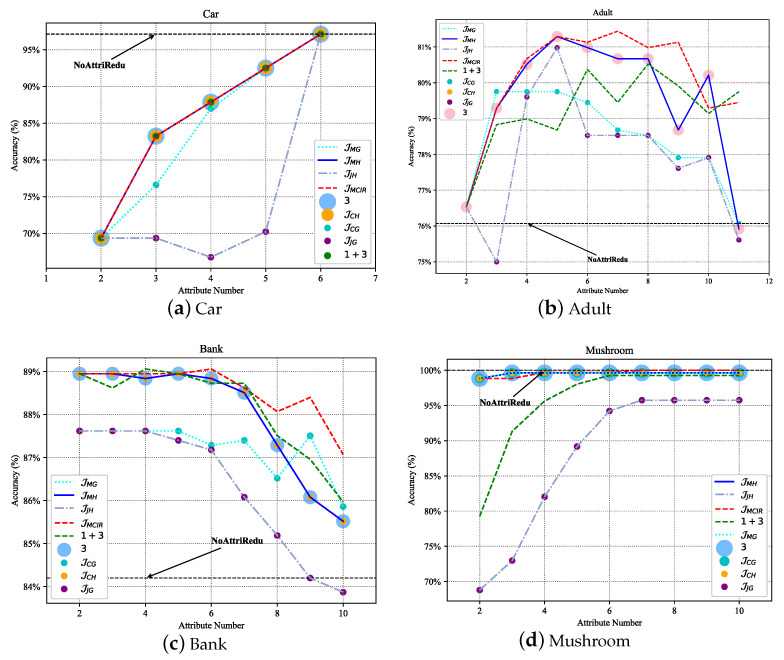
Performance comparison of different attribute reduction algorithms under the MCIR-RGAD algorithm.

**Figure 8 entropy-25-00112-f008:**
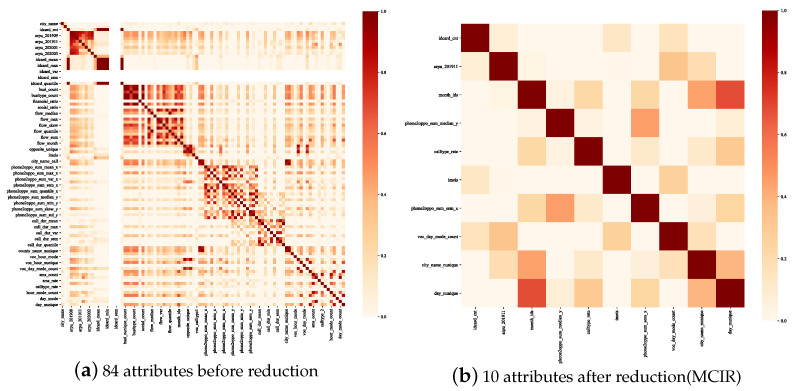
Attribute correlations in the Union dataset.

**Figure 9 entropy-25-00112-f009:**
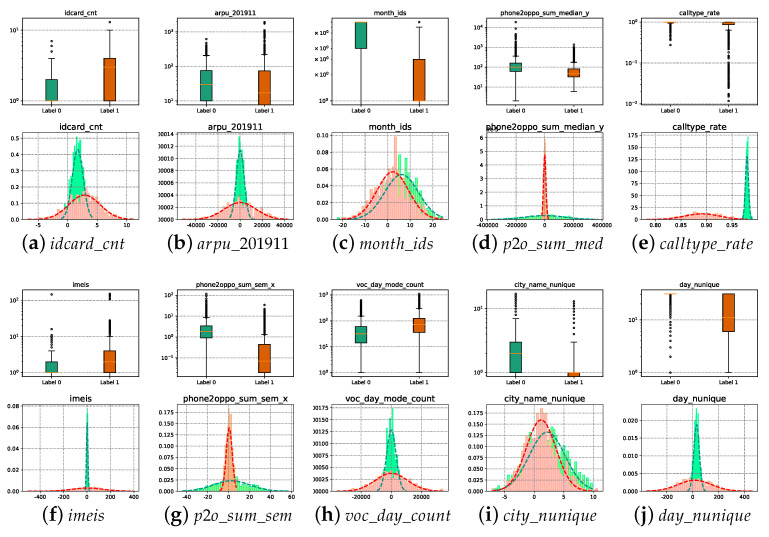
Classification of selected attributes under the MCIR-RGAD algorithm.

**Figure 10 entropy-25-00112-f010:**
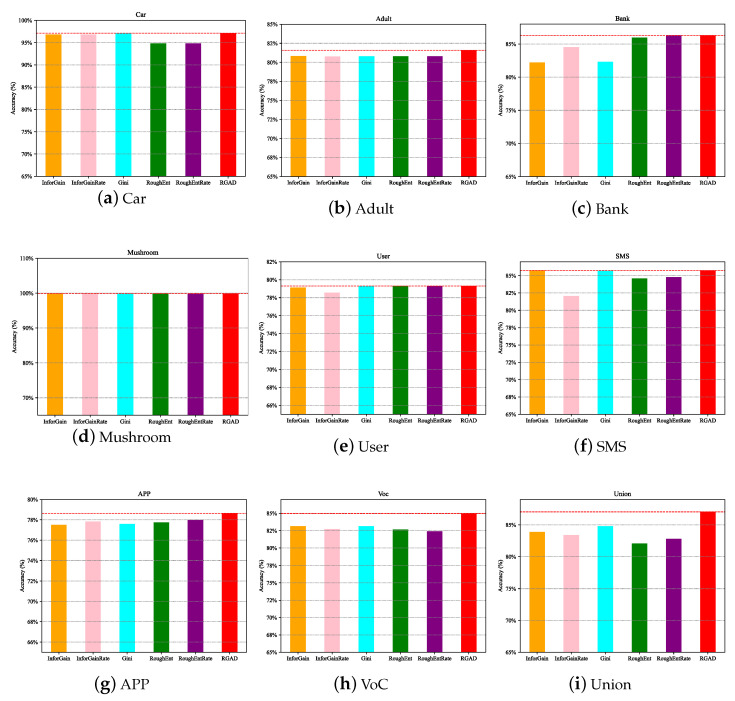
Performance comparison of six classification detection algorithms.

**Figure 11 entropy-25-00112-f011:**
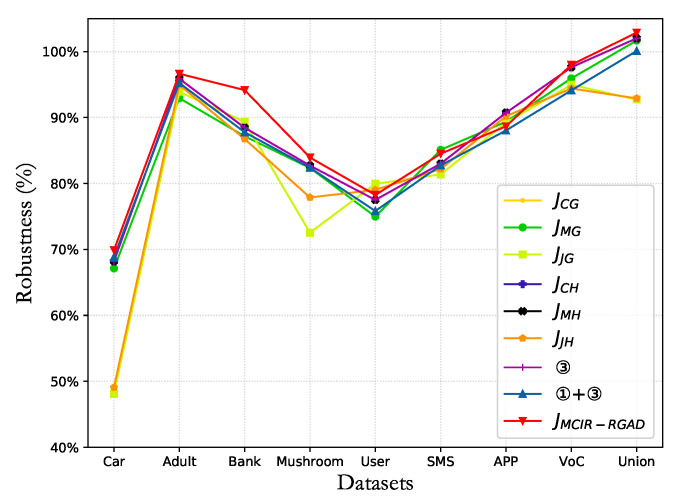
Comparison of the robustness in different datasets.

**Figure 12 entropy-25-00112-f012:**
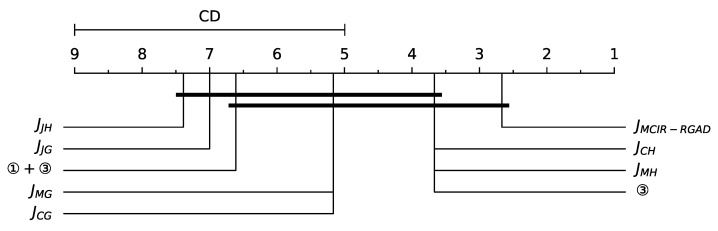
Average ranks diagram comparing the benchmark methods in terms of accuracy.

**Table 1 entropy-25-00112-t001:** An incomplete information system (U,A) about the telecom communication heterogeneous data.

ID	Duration	Amount	Place	Platform	Fraud
$x_{1}$	[60,+∞]	1	Foreign	WeChat	Yes
$x_{2}$	[0,60]	*	Foreign	Telecom	No
$x_{3}$	*	3	*	Telecom	No
$x_{4}$	*	3	Domestic	Telecom	No
$x_{5}$	[60,+∞]	1	Foreign	*	Yes

Note: * means the incomplete information.

**Table 2 entropy-25-00112-t002:** Description of datasets.

Datasets	Sample	Attribute	Source
Car	1728	6 + 1	UCI
Adult	48,842	14 + 1	UCI
Bank	4521	16 + 1	UCI
Mushroom	8124	22 + 1	UCI
USER	6106	11 + 1	Telecom
SMS	6,848,509	11 + 1	Telecom
APP	3,283,602	20 + 1	Telecom
VOC	5,015,430	42 + 1	Telecom
Union	6106	84 + 1	Telecom

**Table 3 entropy-25-00112-t003:** Incomplete information processing of authentic telecom fraud data.

Method	Accuracy	Recall	F1	Current Object	Correct Prediction
Drop	**86.82%**	10.38%	16.42%	4248	3688
Fill 0	78.72%	53.65%	62.10%	6106	4806
Fill 1	78.72%	53.65%	62.10%	6106	4806
Fill Mean	75.86%	48.61%	56.68%	6106	4630
MCB	84.12%	**62.47%**	**71.88%**	**6106**	**5136**

**Table 4 entropy-25-00112-t004:** Comparison of accuracy, computational time, and robustness of attribute reduction in the MCIR-RGAD algorithm.

	Attribute Number		Accuracy(%)		Time(s)		Robustness(%)
Datasets	Before	After	Before	After	Before	After	MCIR-RGAD
Car	6	5	97.11	92.49	0.0373	0.0254	69.91
Adult	14	5	76.07	81.29	0.2765	0.0517	96.64
Bank	16	5	84.20	89.06	0.4618	0.1074	94.16
Mushroom	22	7	100.00	100.00	0.4523	0.1816	83.94
User	11	7	78.81	78.81	0.4090	0.2223	78.26
SMS	11	7	85.75	86.24	0.5548	0.2196	84.51
APP	20	7	77.66	79.54	0.9432	0.3015	88.67
Voc	42	12	84.98	88.05	4.8257	0.4677	97.99
Union	84	10	85.81	89.96	830.2283	0.3520	102.88

**Table 5 entropy-25-00112-t005:** Comparison of classification accuracies of attribute reduction from set view and information view.

	Set Theory View			Information Theory View			Independence		
Datasets	$J_{\mathrm{CG}}$	$J_{\mathrm{MG}}$	$J_{\mathrm{JG}}$	$J_{\mathrm{CH}}$	$J_{\mathrm{MH}}$	$J_{\mathrm{JH}}$	③	① + ③	$J_{\mathrm{MCIR}-\mathrm{RGAD}}$
Car	92.49%	92.49%	70.23%	92.49%	92.49%	70.23%	92.49%	92.49%	**92.49%**
Adult	79.75%	79.75%	80.98%	81.29%	81.29%	80.98%	81.29%	78.68%	**81.29%**
Bank	87.29%	87.29%	87.18%	88.84%	88.84%	87.18%	88.84%	88.73%	**89.06%**
Mushroom	99.63%	99.63%	95.75%	99.63%	99.63%	95.75%	99.63%	99.26%	**100.00**%
User	78.81%	78.81%	78.81%	78.81%	78.81%	78.40%	78.81%	78.81%	**78.81%**
SMS	85.50%	85.50%	80.92%	86.24%	86.24%	80.92%	86.24%	85.59%	**86.24%**
APP	79.87%	79.87%	79.87%	79.54%	79.54%	**79.87**%	79.54%	79.38%	79.54%
VoC	85.15%	85.15%	83.32%	87.97%	87.97%	83.32%	87.97%	82.41%	**88.05%**
Union	88.22%	88.22%	75.44%	88.71%	88.71%	75.68%	88.71%	85.98%	**89.96%**

**Table 6 entropy-25-00112-t006:** Comparison of classification computation time of attribute reduction from set view and information view.

	Set Theory View			Information Theory View			Independence		
Datasets	$J_{\mathrm{CG}}$	$J_{\mathrm{MG}}$	$J_{\mathrm{JG}}$	$J_{\mathrm{CH}}$	$J_{\mathrm{MH}}$	$J_{\mathrm{JH}}$	③	① + ③	$J_{\mathrm{MCIR}-\mathrm{RGAD}}$
Car	0.0280	0.0280	0.0329	0.0270	0.0270	0.0320	0.0270	0.0264	**0.0254**
Adult	0.0689	0.0689	0.0689	0.0569	0.0569	0.0616	0.0569	**0.0471**	0.0517
Bank	0.1737	0.1737	0.1476	0.1715	0.1715	0.1776	0.1715	0.1785	**0.1074**
Mushroom	0.1970	0.1970	0.2817	0.1930	0.1930	0.2213	0.1930	0.1941	**0.1816**
User	0.2559	0.2559	**0.2051**	0.2300	0.2300	0.2112	0.2300	0.2472	0.2223
SMS	0.2042	0.2042	0.2113	0.2405	0.2405	**0.2002**	0.2405	0.2378	0.2196
APP	0.2909	0.2909	0.2943	0.2531	0.2531	0.2712	**0.2531**	0.3134	0.3015
VoC	0.4692	0.4692	**0.4321**	0.5054	0.5054	0.4997	0.5054	0.4570	0.4677
Union	0.2937	0.2937	0.2975	0.3414	0.3414	0.3016	0.3414	**0.2844**	0.3520

**Table 7 entropy-25-00112-t007:** Ranking on 9 datasets for 9 algorithms.

	Ranking Value $r_{i}$ (Accuracy(%))								
Datasets	$J_{\mathrm{CG}}$	$J_{\mathrm{MG}}$	$J_{\mathrm{JG}}$	$J_{\mathrm{CH}}$	$J_{\mathrm{MH}}$	$J_{\mathrm{JH}}$	① + ③	③	$J_{\mathrm{MCIR}-\mathrm{RGAD}}$
Car	4 (92.49)	4 (92.49)	8.5 (70.23)	4 (92.49)	4(92.49)	8.5 (70.23)	4 (92.49)	4 (92.49)	4 (92.49)
Adult	7.5 (79.75)	7.5 (79.75)	5.5 (80.98)	2.5 (81.29)	2.5 (81.29)	5.5 (80.98 )	2.5 (81.29)	9 (78.68)	2.5 (81.29)
Bank	6.5 (87.29)	6.5 (87.29)	8.5 (87.18)	3 (88.84)	3 (88.84)	8.5 (87.18)	3 (88.84)	5 (88.73)	1 (89.06)
Mushroom	4 (99.63)	4 (99.63)	8.5 (95.75)	4 (99.63)	4 (99.63)	8.5 (95.75)	4 (99.63)	7 (99.26)	1 (100.00)
User	4.5 (78.81)	4.5 (78.81)	4.5 (78.81)	4.5 (78.81)	4.5 (78.81)	9 (78.40)	4.5 (78.81)	4.5 (78.81)	4.5 (78.81)
SMS	6.5 (85.50)	6.5 (85.50)	8.5 (80.92)	2.5 (86.24)	2.5 (86.24)	8.5 (80.92)	2.5 (86.24)	5 (85.59)	2.5 (86.24)
APP	2.5 (79.87)	2.5 (79.87)	2.5 (79.87)	6.5 (79.54)	6.5 (79.54)	2.5 (79.87)	6.5 (79.54)	9 (79.38)	6.5 (79.54)
VoC	5.5 (85.15)	5.5 (85.15)	7.5 (83.32)	3 (87.97)	3 (87.97)	7.5 (83.32)	3 (87.97)	9 (82.41)	1 (88.05)
Union	5.5 (88.22)	5.5 (88.22)	9 (75.44)	3 (88.71)	3 (88.71)	8 (75.68)	3 (88.71)	7 (85.98)	1 (89.96)
$R_{ave}^{j}$	5.167	5.167	7	3.667	3.667	7.389	3.667	6.611	2.667

## Data Availability

The datasets used in research are publicly available at https://archive.ics.uci.edu/ml/index.php accessed on 12 April 2022 and https://aistudio.baidu.com/aistudio/datasetdetail/40690 accessed on 12 April 2022. The codes of this research will be made available on request.

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
