# Peer review of "TFD-IIS-CRMCB: Telecom Fraud Detection for Incomplete Information Systems Based on Correlated Relation and Maximal Consistent Block"

_entropy, 2023, doi:10.3390/e25010112_

Round 1

Reviewer 1 Report

In this paper, authors mainly used the correlation of attributes by entropy function to optimize the data quality and then solved the problem of telecommunication fraud detection with incomplete information. the experimental results of authentic telecommunication fraud data and UCI data show that the MCIR-RGAD algorithm provides an effective solution for reducing the computation time, improving the data quality, and processing incomplete data. This article is very valuable. But some issues should be changed.

1. In the introduction, please give the shortcomings of traditional methods. I suggest categorizing and summarizing.

2. The pointed Rough set theory, Information theory should be explained the role in this paper.

3. The criteria JMI[25], CMIM[26], CIFE[27], ICAP[28], ICI[29], MCI[29], etc, authors cited many references, I think one provenance can on behalf of them.

4. Some parameters appear for the first time without explanation, please correct.

5. Future works should be expanded.

6. Some papers should be added to increase the readability.

Xiaowei Wang, Shoulin Yin, Hang Li. A Network Intrusion Detection Method Based on Deep Multi-scale Convolutional Neural Network[J]. International Journal of Wireless Information Networks. 27(4), 503-517, 2020.

Ahmed I M, Kashmoola M Y. CCF Based System Framework In Federated Learning Against Data Poisoning Attacks[J]. Journal of Applied Science and Engineering, 2022, 26(7): 973-981.

Yang J, Yang T, Shi C. Research on fault identification method based on multi-resolution permutation entropy and ABC-SVM[J]. Journal of Applied Science and Engineering, 2021, 25(4): 733-742.

Author Response

Thanks for reviewers' and editors' instant comments.  After carefully studying the comments and advices, we have made corresponding changes in the revised manuscript. 

The documents in the appendix specify the main revisions and supplements to the article.

Do not hesitate to contact us if you have any questions regarding the revised manuscript. We look forward to hearing from you soon.

Season’s Greetings, and all the best in the coming New Year.

Reviewer 2 Report

The paper deals with fraud detection with the telecom market. The paper is interesting and the proposed algorithm seems promising, but the paper must be improved in several ways, which are described next.

First, the paper has some very confusing parts and many typos. Hence, a thorough language review must be carried out. Also, the notation is very confusing, specially in Sections 2 and 3. This prevent the complete understanding of the proposed technique.

In Section 4, the authors must discuss the results shown in all figures, since in the current version of the manuscript, the paper limits to say what is shown in those figures. Also in Section 4, why do the authors use datasets Car, Adult, Bank, and Mushroom from UCI? These are not fraud detection datasets and, from the title of the paper, we expect that the dataset used in the manuscript be related with telecom fraud detection.

Other comments:

- The Introduction must contain a thorough review of telecom fraud detection problem, with previously proposed solutions and challenges;

- The authors should provide a reference to "Vanilla attribute reduction algorithms" on line 43;

- What do the authors mean by "tolerance relation" on line 54?

- What do the authors mean by "time complexity" on line 76? Is it equivalent to "computational complexity"?

- What do the authors mean by "the attributes in the information system can not play a full role" on line 144?

- Figures 2, 4, 7, and 10 are very small;

- The authors must provide the results from the Union dataset on Fig. 5;

- In Fig. 7, the authors must cite all the techniques, whose labels are shown in the legends of the subfigures;

- On line 327, the authors said that the proposed MCIR algortihm achieves better accuracy with fewer attributes. Why? A possible explanation to that behaviour should be provided;

- How do the pdfs shown in Fig. 9 are used to filter out the users as normal or fraudulent?

- In Table 4, the authors should present the results for only telecom fraud related datasets. In this case, it is possible to see that the difference in accuracy between the proposed technique and the other simulated methods are not so big. For instance, for the VoC dataset, a difference between 87.97% and 88.05% is not statistically relevant;

- In Table 6, what does a robustness of 102.88% mean?

- Why do the authors choose k = 0.4 in the robustness index present in (14)?

-

Author Response

(The authors gave the same response as above.)

Reviewer 3 Report

In this paper authors propose methods for telecommunication fraud detection. They proposed an attribute reduction algorithm based on max-correlation and max-independence rate (MCIR) to improve data Another algorithm is proposed a rough-gain anomaly detection algorithm (MCIR-RGAD). The proposed algorithm (MCIR-RGAD) can significantly reduce the computation time but improve accuracy effectively.  But time complexity analysis of algorithms should be done for clear contribution. Also time complexity based comparison to similar methods should be done. In this cade paper would be more valuable.

Author Response

(The authors gave the same response as above.)

Round 2

Reviewer 1 Report

Authors had modified this paper based on comments.

It can be accepted.

Reviewer 3 Report

Authors propose  a method in this paper mainly uses the correlation of attributes by entropy function to optimize the data 3 quality and then solves the problem of telecommunication fraud detection with incomplete information.  Paper satisfied comments provided in this paper